# The Spatial Evolution Law and Water Inrush Mechanism of Mining-Induced Overburden in Shallow and Short Coal Seam Group

**Weidong Pan [1,2], Peng Jiang [1,3,4], Boyang Li [1,2,*], Jianghua Li [3,4] and Yinchao Yang [5]**

[1]  School of Energy and Mining Engineering, China University of Mining & Technology-Beijing, Beijing 100083, China; pwd@cumtb.edu.cn (W.P.); 15201290872@126.com (P.J.)
[2]  Top-Coal Caving Mining Research of Coal Mining Industry, China University of Mining & Technology-Beijing, Beijing 100083, China
[3]  Mine Safety Technology Branch, China Coal Research Institute, Beijing 100013, China; z992471266@163.com
[4]  State Key Laboratory of Coal Mining and Clean Utilization, China Coal Research Institute, Beijing 100013, China
[5]  Kailuan (Group) Co., Ltd., Tangshan 063000, China; zds349636970@163.com
*  Correspondence: 15709545008@163.com

**Abstract:** In order to grasp the overburden evolution law of the shallow and short coal seam group, based on the key bed theory, a mechanical analysis of the longitudinal expansion of mining-induced overburden fractures in the coal seam group was carried out, and the spatial evolution characteristics of mining-induced overburden fractures were simulated by the numerical simulation method. The results show that in the process of working face advancement, with the continuous instability and failure of the overburden, the size and shape of the fracture network are also changed. The repeated mining of the lower coal seam further causes the secondary activation of the upper overburden, which makes the roof fractures of the partially compacted goaf violently move again. The "channel source" and "space source" continue to carry out the process of "generation-expansion-compression-generation-expansion", in combination with pore fracture elastic theory. The water inrush characteristics of the whole coal seam are divided into three "solid-liquid" coupling stages: the original gap seepage stage, the initial water discharge stage of mining fissures and the water inrush stage of fractured rock mass. The steady value of water inflow and its variation characteristics with time are predicted by using the formula of deep well flow in a confined aquifer.

**Keywords:** shallow coal seam; overburden failure; water inrush factor; numerical simulation; water inrush mechanism

## 1. Introduction

The coal seams in the mining areas of central and western China generally have the characteristics of shallow buried depth, thin bedrocks, multi-layer coal and small seam spacing. These characteristics can be collectively referred to as shallow and short coal seam group extract. First, the underlying bedrock thickness of coal seam is small, and the burial depth is no more than 150 m, so it is easy to break through under the influence of coal mining. Secondly, due to the small interval between layers in the short coal seam group, the coal seam mining seams will have a great influence on each other, and the repeated mining will continue to affect the development characteristics of overburdening. Whether there will be a water inrush phenomenon in coal seam mining is mainly affected by the following main control factors: water inrush source, abundance of water source and water pressure, water inrush channel and stagnant water space, which can be referred to as the "four source elements", including the water inrush source, dynamic source, channel source and spatial source. When the "four source elements" constitute a complete relationship, water inrush will occur in the working face. For the problems of overburden failure and

water inrush in coal seam mining under shallow buried conditions, Huang Qingxiang [1–3] studied the ore pressure characteristics of shallow buried coal seams and proposed the theory of "double key layers" in the roof of a near-shallow buried coal seam with large mining height. Jia Mingkui [4] studied the failure height of the overburdening after mining of thin bedrock working face and made clear the conditions of water and sand gushing. For shallow coal seam mining, Yang Dengfeng [5] studied the instability mechanism of overburdened rock under load using catastrophe theory and analyzed the main influencing factors of overburdened rock instability. Fan Limin et al. [6,7] studied the strata control and water conservation mining technology of coal seam mining under the condition of shallow burial in the ecologically fragile region of western China. Liu Quanming [8] studied the buried depth effect of the overburdened migration rule in fully mechanized caving mining under the condition of shallow buried depth and thin bedrock by using the method of a similar simulation experiment. More research focused on single coal seam mining under the condition of impact, for the shallow buried close distance coal seam group mining roof damage rule and the study of water inrush mechanisms is not very perfect, so that under the condition of close distance coal seam group occurrence, a shallow buried coal seam repeated mining group will cause the overburden rock fracture channels and severe changes of the space, causing a water inrush mechanism different from those of single coal seam mining conditions. Therefore, it is of great significance to study the failure law of roof overburdening and the water inrush mechanisms of shallow and short-distance coal seam group mining (Figure 1 research methodology flowchart) and grasp its evolution law to prevent the occurrence of water disasters.

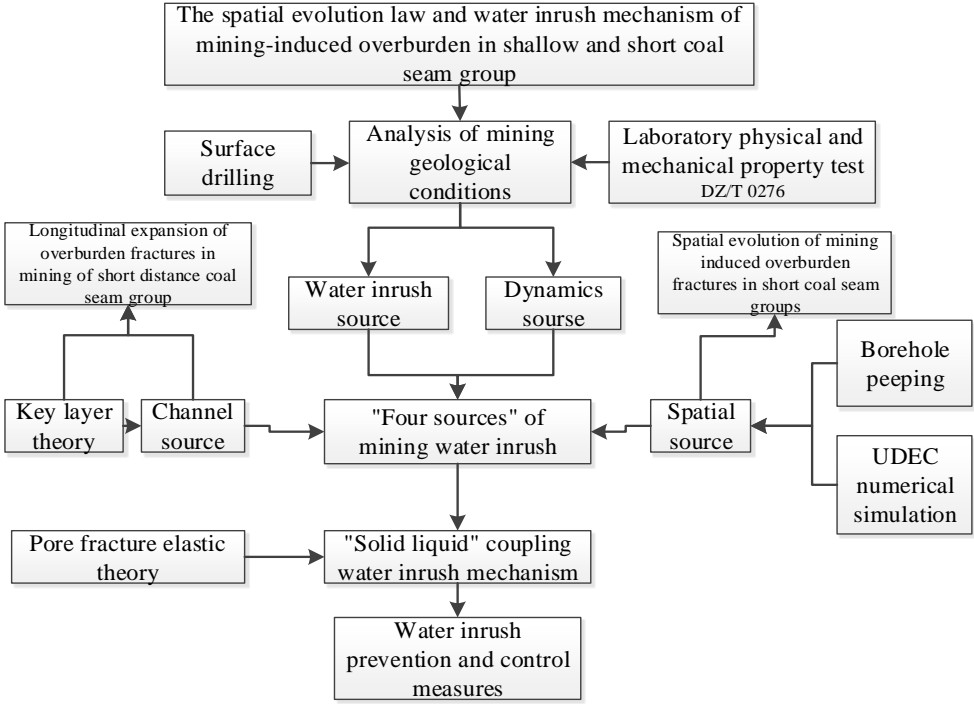

**Figure 1.** Research methodology flowchart.

## 2. Overview of the Study Area

Xialiyuan Coal Mine belongs to the Pingshuo Mining Area. Coal 4 and 9 of the Taiyuan formation are mainly mined in the mine field. The second mine area's first working face of 4 coal is a 40,201 working face. The extension length is 765 m, the dip length is 150 m and the average thickness of the coal seam is 11.2 m with a dip angle of 2°~5°. The structure of the coal seam is simple and stable, the roof contains K3 and K4 sandstone aquifers, the water yield is weak, the average thickness of the bedrock is 100 m, the average thickness of the Quaternary is 30 m, the water yield is medium and the spring water is exposed in

the low-lying part of the ground. The average thickness of a 9 coal seam is 8 m, which, under the 4 coal and the coal seam spacing, is 30.50 m. The fully mechanized top coal caving mining method is adopted in the working face. According to the construction drilling results and the laboratory physical and mechanical property test results of the construction rock samples during the mine field exploration (the test is based on DZ/T 0276 rock physical and mechanical property test specification), the statistical regional borehole columnar characteristics and the physical and mechanical parameters of each coal and rock stratum are shown in Table 1.

**Table 1.** Physical and mechanical parameters of coal and rock.

| Rock Number | Lithology | Thickness (m) | Density (kg/m$^3$) | Elastic Ratio (GPa) | Poisson's Ratio | Cohesion (MPa) | Internal Friction Angle (°) | Uniaxial Tensile Strength (MPa) |
|---|---|---|---|---|---|---|---|---|
| 1 | Loose strata | 30 | 1612 | 0.79 | 0.33 | 0.6 | 45 | 0.43 |
| 2 | Sandy mudstone | 16 | 2364 | 1.56 | 0.35 | 1.37 | 26 | 0.89 |
| 3 | Medium grained sandstone | 4.23 | 2511 | 3.12 | 0.39 | 2.55 | 48 | 2.21 |
| 4 | Siltstone | 4.56 | 2383 | 2.27 | 0.33 | 1.65 | 27 | 1.27 |
| 5 | Sandy mudstone | 12 | 2364 | 1.56 | 0.35 | 1.37 | 26 | 0.89 |
| 6 | Siltstone | 11.5 | 2383 | 2.27 | 0.33 | 1.65 | 27 | 1.27 |
| 7 | Sandy mudstone | 10.4 | 2364 | 1.56 | 0.35 | 1.37 | 26 | 0.89 |
| 8 | Fine grained sandstone | 3.9 | 2420 | 5.1 | 0.29 | 2.36 | 23 | 1.83 |
| 9 | Sandy mudstone | 24.8 | 2364 | 1.56 | 0.35 | 1.37 | 26 | 0.89 |
| 10 | Medium grained sandstone | 9.63 | 2511 | 3.12 | 0.39 | 2.55 | 48 | 2.21 |
| 11 | Sandstone | 2.2 | 2464 | 1.26 | 0.35 | 1.37 | 23 | 0.67 |
| 12 | 4 coal | 11.2 | 1320 | 1.26 | 0.36 | 0.83 | 19 | 0.72 |
| 13 | Sandy mudstone | 12 | 2364 | 1.56 | 0.35 | 1.37 | 26 | 0.89 |
| 14 | Medium grained sandstone | 6.75 | 2511 | 3.12 | 0.39 | 2.55 | 48 | 2.21 |
| 15 | Fine grained sandstone | 4.6 | 2383 | 2.27 | 0.33 | 1.65 | 27 | 1.27 |
| 16 | Sandy mudstone | 7.15 | 2364 | 1.56 | 0.35 | 1.37 | 26 | 0.89 |
| 17 | 9 coal | 8 | 1320 | 1.26 | 0.36 | 0.83 | 19 | 0.72 |
| 18 | Sandy mudstone | 17.6 | 2364 | 1.56 | 0.35 | 1.37 | 26 | 0.89 |

According to the above "four source elements" of water inrush. K3, K4 and the Quaternary aquifer in Xialiyuan coal mine are all water inrush sources. The Quaternary aquifer has a medium water yield and is a stable drastic source. Once qualified channel and spatial sources are produced in coal seam mining, water inrush may occur in the working face. Therefore, by studying the spatial evolution characteristics of mining overburdening in the coal seam, the occurrence process of channel source and spatial source can be obtained, and then the law of water inrush can be mastered and prevented in the future.

### 3. "The Channel Source"—Mechanical Analysis of Longitudinal Propagation of Mining-Induced Overburden Fractures in Short Coal Seam Group

The strata above the direct roof of coal seams are the key strata that control the activities of the whole overburden in the stope or to the surface, so the structural position of the key strata can affect the overall development height of the mining-induced overburden. When the main key strata is located under the overburden and close to the coal seam, and is less than a critical distance, because the compressible rotary space resistance of the lower part of the main key strata is relatively large, the subsidence and rotation of the structure block are larger when the key strata is broken, and the fracture of the key strata can lead to the larger opening of the rock fracture of the strata, which extends to the top of the overburden.

#### 3.1. Key Strata Structure of Overburden in Coal Seam Group

According to the key strata theory [9–11], when in the rock strata above the coal seam $(q_{n+1})_1 < (q_n)_1$, the first hard rock layer and the N + 1 rock layer can both be the key strata, controlling the synchronous fracture and subsidence of other soft rock strata. When $L_{n+1} < L_1$, above the coal seam, only the first hard strata is the key strata, which is a single key stratum. When $L_{n+1} > L_1$, there are at least two key layers in the overburden: the upper one controlling the upper layer to the surface of the hard rock as the main key layer and the lower hard rock as the sub-key layer. According to the judgment criterion of key strata, and referring to the columnar structure of boreholes and the physical and mechanical parameters of strata in Table 1, quantitative analysis and judgment are made on all strata in the working faces of Xialiyuan Coal Mine. The judgment results are as follows: the rock No. 10 (medium-grained sandstone, 9.63 m thick) is the main key layer, and the rock No. 14 (medium-grained sandstone, 6.75 m thick) is the sub-key layer.

In the coal seam No. 4 mining, the main key layer (rock No. 10, referred to as key layer 1) plays a dominant role in controlling the activities of the overburden (Figure 2). When the coal seam No. 4 is exhausted, the coal seam No. 9 is mining, and the sub-key layer (rock layer No. 14, referred to as key layer 2) becomes the main key strata of the overburden of the coal seam No. 9. The key strata structure has changed a single key layer of the upper structure (Figure 3), which can present typical rock pressure characteristics of strata behavior.

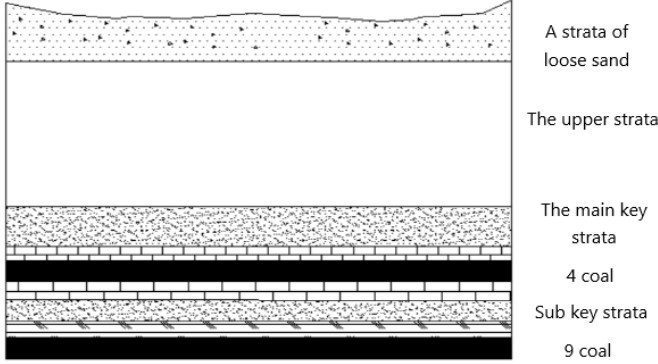

**Figure 2.** Key strata structure of overburden of coal seam group before mining.

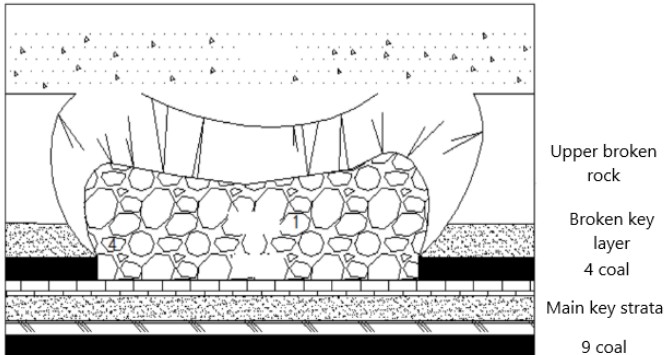

**Figure 3.** Key strata structure of overburden in coal seam No. 4 after mining.

### 3.2. Failure Mechanism of Key Strata in Coal Seam Group

There are two types of overburden strata failure instability [12]: Structure slide instability (S instability, shear failure) and structure deformation instability (R instability, tension bending failure).

#### 3.2.1. Shear Failure Condition

The shear failure condition of overburden is when the working face is pushed to the cracking end of overburden, the middle part of the strata has not cracked and the shear stress of the remaining section has exceeded the ultimate strength. In this case, when there is a small amount of space under the strata, the overburden will be cut off as a whole, resulting in the overburden being cut off as a whole in the goaf [13,14]. According to the rock beam stress analysis and material mechanics [15], the shear failure expression formula is as follows:

$$L_c = \frac{4M_c[\tau]}{3(M + \sum\limits_{i=1}^{n} M_i)\gamma} \tag{1}$$

If $\sum\limits_{i=1}^{n} M_i = 0$, there is:

$$L_c = \frac{4M_c[\tau]}{3M\gamma}$$

where $L_c$—shear distance of rock strata, m: $M_c$—residual thickness of rock beam end after cracking, m; $[\tau]$—Shear strength of rock, MPa; $M$—actual thickness of rock beam, m; $M_i$—thickness of the following layer above the rock beam, m; $\gamma$—average bulk density of rock strata, MPa.

#### 3.2.2. Tensile Bending Failure Condition

The bending failure condition of the overburden on the roof is as follows: With the continuous advance of the stope, the overburden (assumed to be plastic strata) is exposed, and under the action of overlying weak strata and its own gravity, the strata are bent and subsided. The original fixed beam is transformed into a simply supported beam. With the maximum bending moment transferred from the end of the beam to the middle of the beam, the middle part of the beam begins to crack and fall, and finally forms a "pseudo plastic rock beam"; if the height of the lower free space of the rock beam exceeds its allowable settlement value, the rock beam will collapse and the tensile bending failure and instability will occur. The expression formula is:

$$S_0 < S_n = h - \sum\limits_{i=1}^{n-i} M_i(K_A - 1) \tag{2}$$

where $S_0$—Allowable settlement value of "pseudoplastic beam", m; $S_n$—The free space height that allows the movement of the rock, m; $K_A$—Dilatancy coefficient of caved rock; $M$—Thickness of caving rock, m; $h$—Mining height, m.

### 3.2.3. Theoretical Criterion of Overburden Failure form

When $L_c > L_0$, the rock strata are pulled away from the middle.
When $L_c = L_0$, the strata are sheared.
Where $L_c$—Rock shear distance, m; $L_0$—Limit span at first crack of beam end.

### *3.3. Mechanical Analysis of Longitudinal Propagation of Overburden Fracture in Coal Seam Group Mining*

According to the analysis of the strata structure in Xialiyuan coal mine, there are two key strata that play a key role in controlling the whole overburden in the coal seam group working face. The fracture instability characteristics of two key layers, key layer 1 and key layer 2, are analyzed.

In coal seam mining, when the overlying key layer reaches the limit span, it will not collapse immediately, but will form a three hinged arch structure. During the continuous advancement of the mining face, the stability of the three hinged arch structure is mainly controlled by two key blocks. The interaction of the two key blocks presents two failure forms: sliding instability (s) and deformation instability (R).

### 3.3.1. Failure Judgment of Key Layer "S-R"

Judgment of Key Stratum Sliding Instability (S)

After the initial fracture of the key strata, the two rock blocks form an arch equilibrium state (Figure 4). The horizontal extrusion force $T$ can be considered as the action point at a/2 (Figure 5), the load borne by the key strata is $p$ and the length and height of the rock block are $l$ and $h$, respectively. The moment balance is obtained for point $O$ to obtain the balance curve formula without sliding instability:

$$i \le \tan \varphi + \sin \alpha \tag{3}$$

$$\sin \alpha = [M - \sum h(k_p - 1)]/l \tag{4}$$

where $i$—Fracture degree of rock block, ratio of height $H$ to length $L$ of fracture block; $\tan\varphi$—The friction coefficient between rocks is 0.3; $\alpha$—Block rotation angle, $M$—Mining height; $H$—Height of rock strata falling; $K_P$—Breaking expansion coefficient, take 1.5; $L$—Length of fractured rock block.

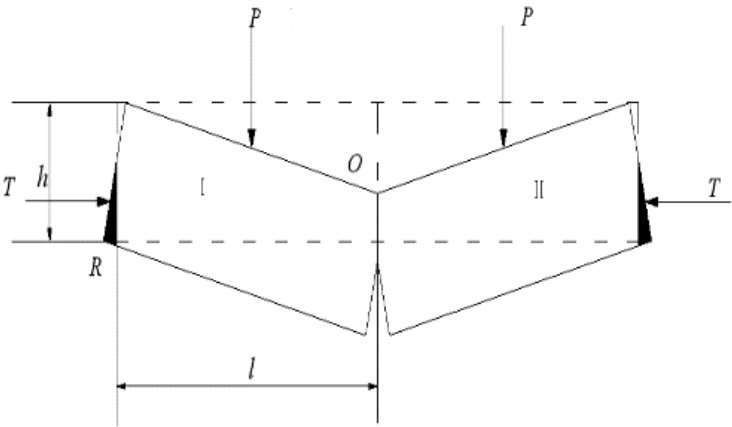

**Figure 4.** Schematic diagram of three hinged arch structure.

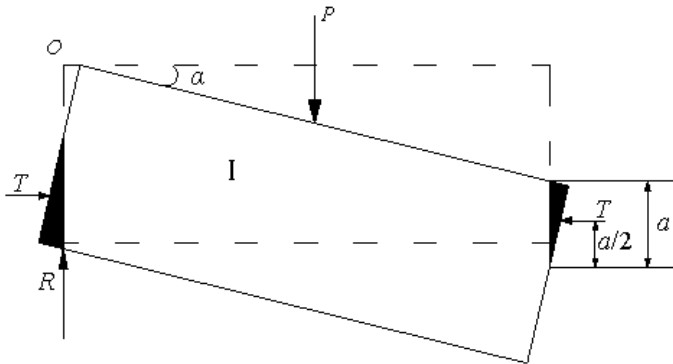

**Figure 5.** Schematic diagram of stress analysis of key blocks.

Rotation Instability of Key Layer (R)

According to the relationship between T/P and the rotation angle $\alpha$, when the fracture degree is constant, the horizontal thrust T between the key blocks increases with the increase of the rotation angle, which will lead to the crushing of the corner blocks and deformation and rotation instability. In order to avoid the deformation and instability of the key blocks, the following conditions should be met:

$$\frac{T}{a} \leq \eta\sigma_c \tag{5}$$

where $\sigma_p = T/a$ —The extrusion stress formed at the bite; $\eta$—The extrusion coefficient at the end angle of the key block, $\eta\sigma_c$ is the extrusion strength, $\eta$ is 0.3.

According to the above formula, the judgment curve formula of rock block without rotary instability is:

$$i \geq \sin\alpha / (1 - \sqrt{\frac{nk\eta}{3}}) \tag{6}$$

where $n$—Ratio of compressive strength to tensile strength of rock; $k$—Calculated as simply supported beam, $k = 0.3$; $\eta$—Ratio of extruding strength to compressive strength of rock block, $\eta$ is 0.3.

According to the above analysis, the instability form of rock block mainly depends on the fracture degree and rotation angle of the block.

Judgment of Instability form of Key Strata

① Judge the instability form of key strata 1 and substitute the relevant parameters of the above judgment formula: $\tan\varphi$ is 0.3; $H$ is 9.63 m; $M$: The mining height of coal seam No. 4 is 11.2 m; $h$ is 14 m; $k_p$ is 1.5; $l$ is 12.14 m.

Calculated $i = 0.793$, $\tan\varphi + \sin\alpha = 0.646$, $i > \tan\varphi + \sin\alpha$, Therefore, key strata 1 will slip and lose stability.

$$\sin\alpha / (1 - \sqrt{\frac{nk\eta}{3}}) = 0.544,$$

$$i \geq \sin\alpha / (1 - \sqrt{\frac{nk\eta}{3}})$$

Therefore, there will be no rotary instability in key strata 1.

② Judge the instability form of key strata 2, and substitute the relevant parameters of the above judgment formula: $\tan\varphi$ is 0.3; $H$ is 6.75 m; $M$: The mining height of coal seam No. 9 is 8 m; $h$ is 11.5 m; $k_p$ is 1.5; $l$ is 10.3 m.

Calculated $i = 0.655$, $\tan\varphi + \sin\alpha = 0.518$, $I > \tan\varphi + \sin\alpha$, Therefore, key strata 2 will slip and lose stability.

$$\sin\alpha / (1 - \sqrt{\frac{nk\eta}{3}}) = 0.334$$

$$i \geq \sin\alpha / (1 - \sqrt{\frac{nk\eta}{3}})$$

Therefore, there will be no rotary instability in key strata 2.

### 3.3.2. Longitudinal Development Characteristics of Mining-Induced Overburden Fractures in Close Coal Seam Groups

Through the instability of key strata fracture mechanism analysis, the mining face in the working face end composite beams above will produce open fractures, and gradually, from the bottom up, when the rotary rock fracture gradually increases, the main roof structure sliding instability occurs, being exposed for the first time to pressure, formatting a water bursting fracture source "channel", pushing the continuous mining face forward. As a result, the key rocks in the overburden roof break periodically quickly, and the longitudinal fractures in the overburden rock continue to develop upward through the wave base rock. During the mining of coal seam No. 9 in the lower group, the fracture instability of the key strata will directly lead to the fracture of the roof rock in the goaf of coal seam No. 4, resulting in the secondary activation of the upper overburden, resulting in the violent activity of the roof cracks in the partially compacted goaf again, as well as the expansion of the crack channel and the secondary development of overburden cracks. Longitudinal development model of overlying rock fissure in close coal seam mining (see Figure 6).

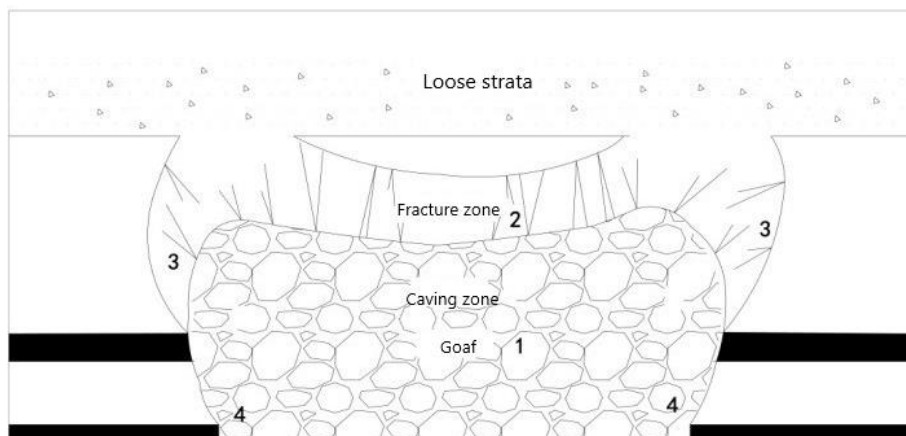

**Figure 6.** Longitudinal development model of overlying rock fissure in close coal seam mining. 1. Tensile failure zone. 2. Tensile fracture zone. 3. Shear failure zone. 4. Tensile shear failure zone.

## 4. "Spatial Source"—Spatial Evolution Characteristics of Mining-Induced Overburden Fractures in Short Distance Coal Seam

The structure of goaf and overburdening after mining can be divided into relatively complete layered structure, massive structure and broken structure, according to the degree of fragmentation and form, and the overall combination mode is closely related to the size of the water accumulation space. In the original strata of mining, the water accumulation space mainly exists in rock micro fractures. After coal mining, the violent development of overburden fractures leads to the rapid expansion of the water accumulation space, and rocks in the overburden fracture zone are massive structure rocks (see Figure 7), so the water accumulation space is mainly distributed in the pore fracture channel. The rock mass in the caving zone, which has been seriously damaged, presents a broken structure, and the overburden space is fully developed, as is the main water accumulation area. The spatial evolution characteristics of mining-induced overburden fractures are simulated by numerical simulation.

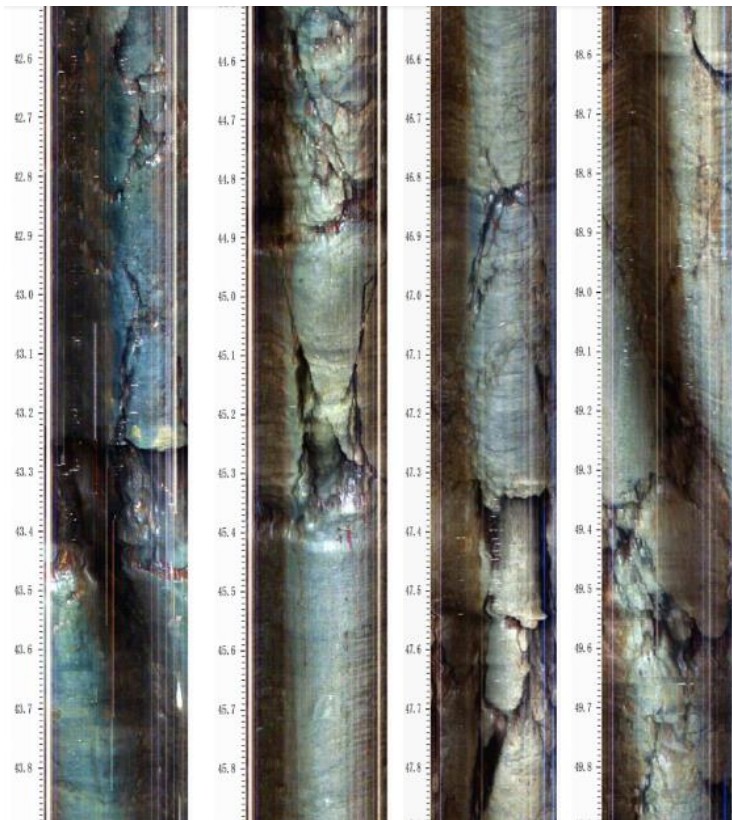

**Figure 7.** Borehole television map of fracture development in overburden after mining.

The following 40,201 working face area of Xialiyuan Coal Mine is taken as an example. According to the strata distribution and physical and mechanical parameters of coal and rock in Table 1, UDEC numerical simulation software is used to establish a numerical analysis model to simulate the spatial variation characteristics of overburdened fractures in the process of coal seam group mining (see Figures 8 and 9) [16].

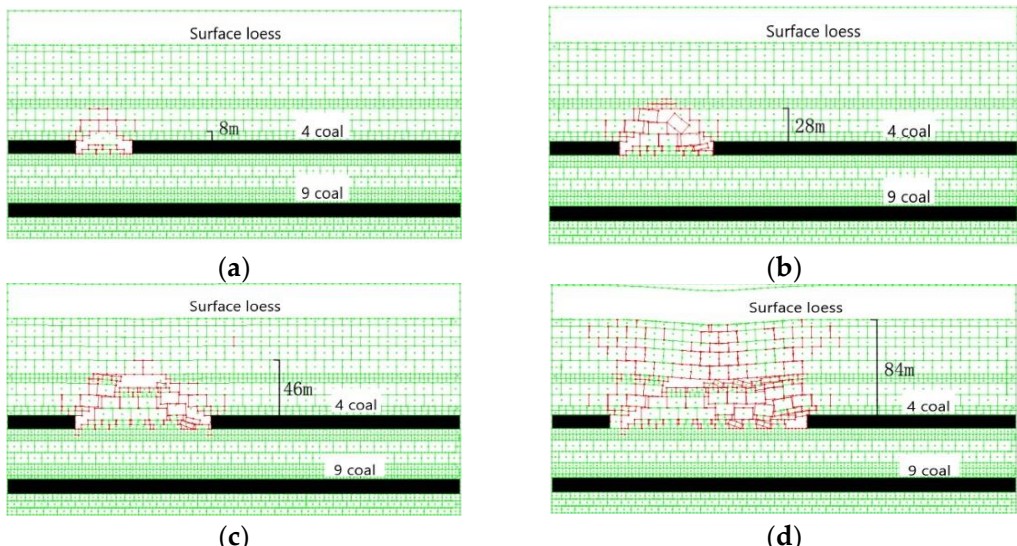

**Figure 8.** Spatial variation characteristics of overburden in single coal seam of coal seam No. 4. (**a**) Working face advance 20 m. (**b**) Working face advance 40 m. (**c**) Working face advance 100 m. (**d**) Working face advance 160 m.

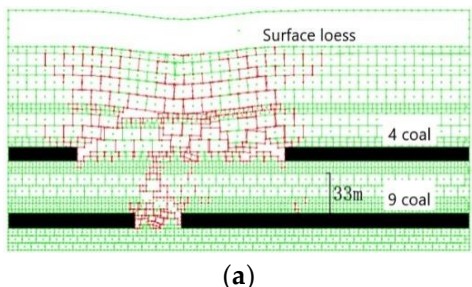
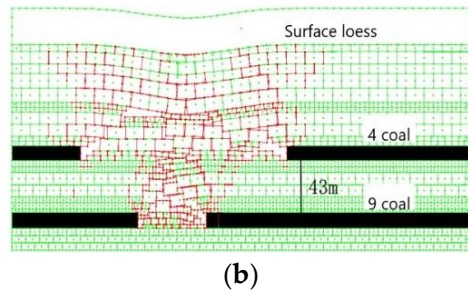

(**a**)          (**b**)

**Figure 9.** Spatial variation characteristics of overburden under the influence of repeated mining of coal seam No. 9. (**a**) Working face advance 40 m. (**b**) Working face advance 50 m.

In the mining of single coal seams, when the working face advances 20 m, the direct roof collapses and begins to form a water storage space. The overhanging area of the main roof is large, local cracks appear in the lower part of the main roof and fracture instability begins to appear. At this time, after the direct roof collapses, the main roof does not completely collapse under the support of coal pillars on both sides of the goaf, so the goaf is not fully filled, and there is a large space on its left and right. When the aquifer is connected with the goaf, this area becomes the main water accumulation space. When it is excavated to 40 m, initial breaking of main roof occurs. Therefore, it can be inferred that the initial pressure step distance is between 35 and 40 m. In terms of fracture form, due to the large lower space after coal caving, the whole stratification under the main roof first cuts off, the cracks continue to develop in the main roof top space and the water accumulation space keeps increasing. When the working face reaches 160 m, spatial fractures in overburdened rock are basically fully developed, and overburdened rock fractures have spread to the surface, At this time, the middle zone of the goaf fully sinks, and the rock in the caving fracture zone is compressed due to the vertical compressive stress. The water source cannot gather in this area, mainly in the incompletely compacted cracks.

In lower coal seam No. 9, repeated mining and the advancement of the working face to 40 m have caused the overburden fracture to spread to the 4 coal floor; when the working face advances to 50 m, the overburden fracture breaks completely through the 4 coal floor, which occurs as a result of repeated 9 coal mining and has carried on the secondary development of overburdening, which increases in a crack in the channel, increasing the water space, making the risk of water inrush bigger.

In the process of the working face advancing, with the continuous instability and failure of overburdening, the space, size and shape of the fracture network changes accordingly. The general evolution process of the fracture network space is compaction and generation at the same time.

## 5. "Solid-Liquid" Coupling Water Inrush Mechanism in Coal Seam Mining

As mentioned above, water inrush in coal seam mining is the interaction of water inrush source, dynamic source, channel source and spatial source. The sudden water source and dynamic source are the original conditions that existed before the mining of the coal seam, that is, there is a strong aquifer. The mining of the coal seam leads to the change of the structure of the upper overburden, and then the channel source and spatial source are generated. Overburden fracture is the channel of water seepage, migration and water storage space. The seepage disaster process of rock strata is the result of the coupling effect of a water-bearing body and consolidated rock mass [17]. According to the deformation and instability process of roof strata, the elastic theory of pore-fissure is referred to [18–21], and the water inrush process of coal seam mining is divided into three "solid-liquid" coupling action stages:

### 5.1. Original Gap Seepage Stage

Before the coal seam is mined, there is an original fissure in the overburden mass. At this time, the water channel and water accumulation space are insufficient, and the aquifer carries out low-speed seepage under the action of water flow. The solid-liquid coupling equation is as follows:

$$\begin{cases} Gu_{i,jj} + (\lambda + G)u_{k,ki} + \sum_{m=1}^{2} \alpha_m p_{m,i} = 0 \\ -\frac{\mu}{k} p_{m,kk} = \alpha_m \varepsilon_{kk} - c^* p_m \pm \Gamma(\Delta p) \end{cases} \tag{7}$$

where $m$:$m = 1$ represents rock foundation, $m = 2$ represents rock gap; $G$ is the shear modulus, Pa; $\lambda$ is the lame constant; $u_{i,jj}$, $u_{k,ki}$ is displacement, m; $\alpha_m$ is Biot constant; $p_{m,i}$ is the fluid pressure, Pa; $k$ is the equivalent permeability value or the average permeability of the overall system, m$^2$; $\mu$ is the dynamic viscosity of fluid, m$^2$/s; $c^*$ is lumped compressibility; $\Gamma$Is the fluid exchange rate caused by differential pressure $\Delta P$.

### 5.2. Initial Water Discharge Stage of Mining-Induced Fracture

During coal seam mining, under the influence of mining, the cracks of overburden rock mass expand and gradually form connectivity; under the action of the water pressure of the aquifer, the water from an aquifer can move down into the goaf through the cracks in the overburden, which is manifested as seepage in the roof of the working face. The solid-liquid coupling equation satisfied by the seepage of mining fractured rock mass in this process is:

$$\begin{cases} Gu_{i,jj} + (\lambda + G)u_{k,ki} + \sum_{m=1}^{2} \alpha_m p_{m,i} = 0 \\ -\frac{k_m}{\mu} p_{m,kk} = \alpha_m \varepsilon_{kk} - c_m^* p_m \pm \Gamma(\Delta p) \end{cases} \tag{8}$$

where $k_m$ is permeability of m phase, m$^2$.

### 5.3. Water Inrush Disaster Stage of Fractured Rock Mass

With the advance of coal seam mining, the overburden fractures will further expand until the rock mass structure is damaged and collapses. At this time, the aquifer water has a sufficient migration channel and water storage space, and breaks into the goaf in a large area in the form of water inrush at the working face. The solid-liquid coupling equation satisfied by the seepage of broken rock mass in this process is:

$$\begin{cases} Gu_{i,jj} + (\lambda + G)u_{k,ki} + \sum_{m=1}^{3} \alpha_m p_{m,i} = 0 \\ -\frac{k_{13}}{\mu} p_{1,kk} = \alpha_1 \varepsilon_{kk} - c_1^* p_1 + \Gamma_{12}(p_2 - p_1) + \Gamma_{13}(p_3 - p_1) \\ -\frac{k_{13}}{\mu} p_{2,kk} = \alpha_2 \varepsilon_{kk} - c_2^* p_2 + \Gamma_{21}(p_1 - p_2) + \Gamma_{23}(p_2 - p_3) \\ -\frac{k_3}{\mu} p_{3,kk} = \alpha_3 \varepsilon_{kk} - c_3^* p_3 + \Gamma_{31}(p_1 - p_3) + \Gamma_{32}(p_2 - p_3) \end{cases} \tag{9}$$

where $m = 1$, 2 and 3 represent pores, fissures and fractures, respectively; $k_1$, $k_2$ and $k_3$ are pores, fractures and fracture permeability, m$^2$; $k_{13}$ is the average permeability of rock foundation and fracture, m$^2$; $\Gamma_{ij}$ is the fluid exchange rate between phase I and phase J, and it is assumed that there is interstitial flow caused by internal pressure difference between the two phases; others conform to the same meaning.

## 6. Prevention and Control Measures of Water Inrush in Mining Face

### 6.1. Forecast of Water Inrush and Inflow in Working Face

The accurate prediction of water inflow plays an important role in preventing and controlling the occurrence and control of water disasters and can guide mines to formulate prevention and control measures. According to the characteristics of water inrush, the inflow formula of a confined aquifer with a constant depth is used to predict the mine

water inflow [22–24], and the stable value of inflow and its variation characteristics with time are predicted.

### 6.1.1. Mathematical Model

In the early stage of coal seam mining, because the drainage time was not long enough, the falling funnel kept expanding and did not reach the stable flow state. However, the water level has been reduced to the roof of the coal seam, so the mathematical model of the water flow movement should be a definite solution of the variable flow with constant depth. Its mathematical model is

$$\begin{cases} \frac{1}{r}\frac{\partial}{\partial r}\left(r\frac{\partial s}{\partial r}\right) = \frac{S}{T}\frac{\partial s}{\partial t} & t > 0, 0 < r < \infty \\ s(r,0) = 0 & 0 < r < \infty \\ s(\infty,0) = 0 & t > 0 \\ s(0,t) = s_w & t > 0 \end{cases} \tag{10}$$

This mathematical model is obtained by Laplace transform and Darcy's law, and is:

$$Q = 2\pi T s_w G(\lambda) \tag{11}$$

where $s_w$ is the drawdown in the well; $\lambda = \frac{Tt}{r_w^2 S}$ is dimensionless time; $G(\lambda)$ is the flow function of constant drawdown well flow in confined aquifer without overflow recharge.

### 6.1.2. Forecast of Water Inflow

In order to obtain the process Q(t) of water inflow with time, different time points $t_1$, $t_2$, ... and $t_n$ take in (6.17); the corresponding values at different time points can be obtained as G ($\lambda$1), G ($\lambda$2), ... and G ($\lambda$n) and $Q_1$, $Q_2$, ... and $Q_n$. The value of G ($\lambda$) is shown in Table 2.

**Table 2.** G ($\lambda$) values.

| $\lambda$ N | $N \times 10^{-4}$ | $N \times 10^{-3}$ | $N \times 10^{-2}$ | $N \times 10^{-1}$ | $N \times 10^{0}$ | $N \times 10^{1}$ | $N \times 10^{2}$ | $N \times 10^{3}$ |
|---|---|---|---|---|---|---|---|---|
| 1 | 59.6 | 18.34 | 6.13 | 2.249 | 0.985 | 0.534 | 0.346 | 0.251 |
| 2 | 40.4 | 13.11 | 4.47 | 1.716 | 0.803 | 0.461 | 0.311 | 0.232 |
| 3 | 33.1 | 10.79 | 3.74 | 1.477 | 0.719 | 0.427 | 0.294 | 0.222 |
| 4 | 28.7 | 9.41 | 3.3 | 1.333 | 0.667 | 0.405 | 0.283 | 0.215 |
| 5 | 25.7 | 8.47 | 3 | 1.234 | 0.63 | 0.389 | 0.274 | 0.21 |
| 6 | 23.5 | 7.77 | 2.78 | 1.16 | 0.602 | 0.377 | 0.268 | 0.206 |
| 7 | 21.8 | 7.23 | 2.6 | 1.103 | 0.58 | 0.367 | 0.263 | 0.203 |
| 8 | 20.4 | 6.79 | 2.46 | 1.057 | 0.562 | 0.359 | 0.258 | 0.2 |
| 9 | 19.3 | 6.43 | 2.35 | 1.018 | 0.547 | 0.352 | 0.254 | 0.198 |
| 10 | 18.3 | 6.13 | 2.25 | 0.985 | 0.534 | 0.346 | 0.251 | 0.196 |
| N | $N \times 10^{4}$ | $N \times 10^{5}$ | $N \times 10^{6}$ | $N \times 10^{7}$ | $N \times 10^{8}$ | $N \times 10^{9}$ | $N \times 10^{10}$ | $N \times 10^{11}$ |
| 1 | 0.1964 | 0.1608 | 0.136 | 0.1177 | 0.1037 | 0.0927 | 0.0838 | 0.0764 |
| 2 | 0.1841 | 0.1542 | 0.1299 | 0.1131 | 0.1002 | 0.0899 | 0.0814 | 0.0744 |
| 3 | 0.1777 | 0.1524 | 0.1299 | 0.1131 | 0.1002 | 0.0899 | 0.0814 | 0.0744 |
| 4 | 0.1733 | 0.1449 | 0.1244 | 0.1089 | 0.0968 | 0.0872 | 0.0792 | 0.0726 |
| 5 | 0.1701 | 0.1426 | 0.1227 | 0.1076 | 0.0958 | 0.0864 | 0.0785 | 0.072 |
| 6 | 0.1675 | 0.1441 | 0.1213 | 0.1066 | 0.095 | 0.0857 | 0.0779 | 0.0716 |
| 7 | 0.1654 | 0.1393 | 0.1202 | 0.1057 | 0.0943 | 0.0851 | 0.0774 | 0.0712 |
| 8 | 0.1636 | 0.138 | 0.1192 | 0.0105 | 0.0937 | 0.0846 | 0.077 | 0.0709 |
| 9 | 0.1621 | 0.1369 | 0.1184 | 0.1043 | 0.0932 | 0.0842 | 0.0767 | 0.0706 |
| 10 | 0.1608 | 0.136 | 0.1177 | 0.1037 | 0.0927 | 0.0838 | 0.0764 | 0.0704 |

Water Inflow of Coal Seam No. 4

The calculation parameters of water inflow of coal seam No. 4 are shown in Table 3, and the value of time t is shown in Table 4.

**Table 3.** Coal No. 4 water inflow time value table.

|  | $t_1$ | $t_2$ | $t_3$ | $t_4$ | $t_5$ | $t_6$ | $t_7$ | $t_8$ | $t_9$ | $t_{10}$ |
|---|---|---|---|---|---|---|---|---|---|---|
| Time (d) | 10 | 30 | 50 | 100 | 150 | 200 | 300 | 400 | 500 | 600 |

**Table 4.** Calculation of water inflow of unsteady flow coal No. 4.

| Time t (d) | λ | G (λ) | Inflow Q (m³/h) |
|---|---|---|---|
| 10 | 0.00016 | 48.08 | 194.8 |
| 30 | 0.00049 | 25.40 | 102.9 |
| 50 | 0.00082 | 20.18 | 81.8 |
| 100 | 0.00164 | 15.20 | 61.6 |
| 150 | 0.00246 | 11.95 | 48.4 |
| 200 | 0.00328 | 10.43 | 42.3 |
| 300 | 0.00492 | 7.84 | 31.8 |
| 400 | 0.00656 | 7.50 | 30.4 |
| 500 | 0.00821 | 6.86 | 27.8 |

Bring the parameter values in Table 3 into formula 11 and take values according to Table 2 to obtain (Table 4 and Figure 10).

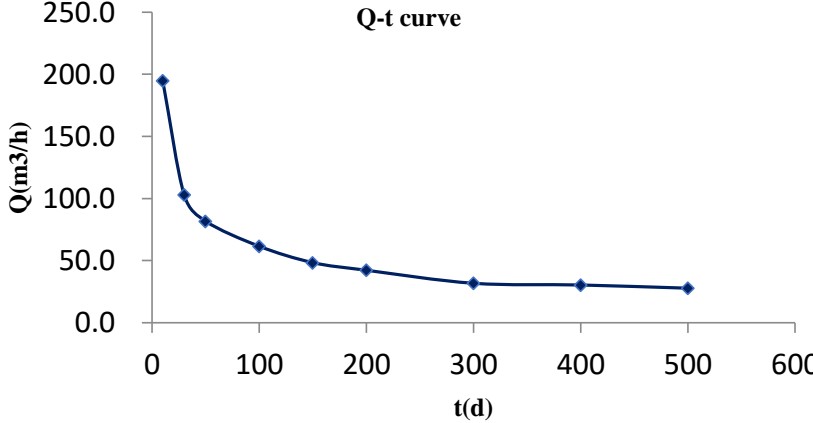

**Figure 10.** Coal No. 4 inflow process line.

It can be seen from Figure 10 that after about 300 days of drainage, the water inflow of coal 4 is basically stable, and the flow is 27.8 m³/h.

Water Inflow of Coal Seam No. 9

The calculation parameters of the water inflow of coal seam No. 9 are shown in Table 2, and the value of time t is shown in Table 5.

**Table 5.** Coal No. 9 water inflow time value table.

|  | $t_1$ | $t_2$ | $t_3$ | $t_4$ | $t_5$ | $t_6$ | $t_7$ | $t_8$ | $t_9$ | $t_{10}$ |
|---|---|---|---|---|---|---|---|---|---|---|
| Time (d) | 10 | 30 | 50 | 100 | 200 | 300 | 400 | 500 | 600 | 700 |

Bring the parameter values in Table 5 into formula 11 and take values according to Table 2 to obtain (Table 6 and Figure 11).

**Table 6.** Calculation of water inflow of unsteady flow coal No. 9.

| Time t (d) | λ | G (λ) | Inflow Q (m³/h) |
|---|---|---|---|
| 10 | 0.00041 | 28.40 | 775.3 |
| 30 | 0.00122 | 17.29 | 472.0 |
| 50 | 0.00204 | 13.11 | 357.9 |
| 100 | 0.00408 | 9.41 | 256.9 |
| 200 | 0.00817 | 6.72 | 183.5 |
| 300 | 0.01225 | 5.80 | 158.3 |
| 400 | 0.01633 | 5.13 | 140.0 |
| 500 | 0.02041 | 4.40 | 120.1 |
| 600 | 0.02450 | 4.11 | 112.2 |
| 700 | 0.02858 | 3.85 | 105.1 |

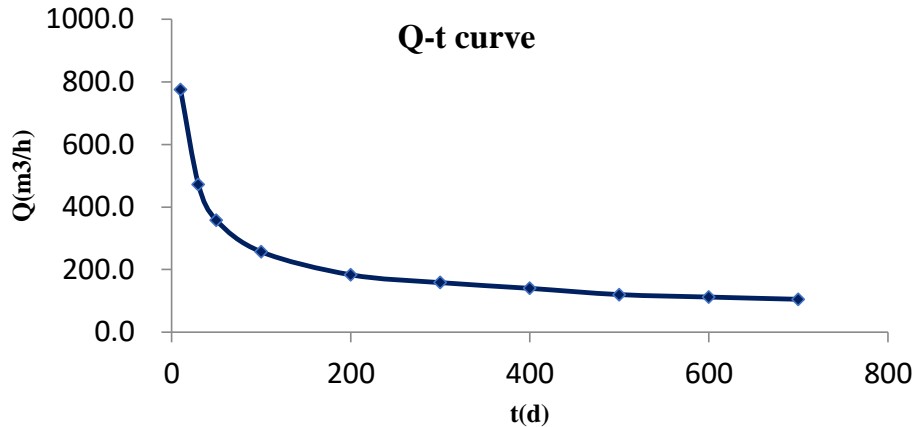

**Figure 11.** Coal No. 9 inflow process line.

It can be seen from Figure 11 that after about 500 days of drainage, the water inflow of coal 9 is basically stable, and the flow is 105.10 m³/h.

*6.2. Water Inrush Prevention and Control Measures*

(1) The water-rich area of the working face should be appropriately accelerated. On the one hand, the processing speed can shorten the overburden dynamic deformation process time; the better the overburden compaction effect is, the more the compression of "spatial source". On the other hand, the accelerating speed can shorten the development time of the overlying rock fissures and can quickly close, preventing the development of a "channel source".

(2) Drainage of water-rich aquifers in the upper working face before mining, as well as drilling and drainage in advance and drainage and pressure reduction to reduce the "water inrush source" and eliminate the "dynamic source".

(3) Thickness limited mining or local filling in the water-rich area of the working face. This method can restrain the development of longitudinal fractures in working face.

## 7. Conclusions

(1)  Water inrush from coal seam mining consists of "four source elements", which are water inrush source, dynamic source, channel source and spatial source, The water inrush source and dynamic source are the original conditions existing before coal seam mining; the mining of the coal seam changes the upper overburden structure and then produces the channel source and spatial source; when the "four source elements" constitute a complete relationship, water inrush will occur in the working face.

(2)  Using the key strata theory, this paper carries out a mechanical analysis of the longitudinal expansion of overburden fractures during mining of short distance coal seam groups and grasps the development and evolution characteristics of a "channel source". Through mechanical model analysis, the two key strata that play a key role in controlling the whole overburden in the mining of coal seam groups in the study area have the form of shear failure (sliding instability) during mining.

(3)  The UDEC numerical simulation method is used to simulate the spatial evolution characteristics of mining-induced overburden fractures. The results show that in the process of the working face advancing, with the continuous instability and failure of overburden, the spatial size and shape of fracture network are also changing, and the general evolution process is as follows: generation-compaction and generation-synchronous action.

(4)  The seepage catastrophe process of rock stratum is the result of the intercoupling of a water bearing body and a consolidated rock mass. According to the deformation and instability process of roof strata and the theory of pore-fracture elasticity, the water inrush process of coal seam mining is divided into three "solid-liquid" coupling action stages: the original gap seepage stage, the initial water discharge stage of a mining-induced fracture and the water inrush disaster stage of a fractured rock mass.

**Author Contributions:** Formal analysis, Y.Y.; Methodology, J.L.; Project administration, W.P.; Resources, P.J.; Writing—review & editing, B.L. All authors have read and agreed to the published version of the manuscript.

**Funding:** This research was funded by [supported by the Fundamental Research Funds for the Central Universities] grant number [2021YQNY10]; This research was funded by [supported by the Fundamental Research Funds for the Central Universities] grant number [2021YJSNY03]; This research was funded by [Technology innovation fund of coal academy of sciences] grant number [280410027-ZC].

**Institutional Review Board Statement:** Not applicable.

**Informed Consent Statement:** Not applicable.

**Conflicts of Interest:** The authors declare no conflict of interest.

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
