# Peer review of "The Spatial Evolution Law and Water Inrush Mechanism of Mining-Induced Overburden in Shallow and Short Coal Seam Group"

_sustainability, doi:10.3390/su14095320_

Round 1

Reviewer 1 Report

Comments: Sustainability (Manuscript ID 661147)

The authors made lots of effort for the research works on “The spatial evolution law and water inrush mechanism of mining-induced overburden in shallow and short coal seam group”. This topic is interesting and well written.  Unfortunately, the authors need to revise the current manuscript to publish it in a peer-reviewed journal. Authors should be adopted the following comments to improve the quality of the manuscript for further consideration.

  1. Abstract: The reviewer would like to suggest revising the abstract by adding the main research findings and novelty of the research; not focusing on introductory information. Also need to re-write the abstract with simple sentences for a better understanding of the readers.   
  2. Additionally, the authors need to address knowledge gaps on the studied research topic to depict the importance of the study.
  3. Need more explanation of all results from Table 1, how authors got the results in terms of experimental procedures.  

Author Response

Point 1: Abstract: The reviewer would like to suggest revising the abstract by adding the main research findings and novelty of the research; not focusing on introductory information. Also need to re-write the abstract with simple sentences for a better understanding of the readers.

Response 1: We appreciate the considerate comment. We changed the abstract and add the content of the current research situation to the paper, and reflect the importance of the research.

In order to grasp the overburden evolution law of the shallow and short coal seam group, based on the key bed theory, the mechanical analysis of the longitudinal expansion of mining-induced overburden fractures in the coal seam group was carried out, and the spatial evolution characteristics of mining-induced overburden fractures were simulated by numerical simulation method. The results shown that in the process of working face advancement, with the continuous instability and failure of the overburden,the size and shape of fracture network are also changed. The repeated mining of the lower coal seam further causes the secondary activation of the upper overburden, which makes the roof fractures of the partially compacted goaf violently move again. The "channel source" and "space source" continue to carry out the process of "generation-expansion-compression-generation-expansion", in combination with pore fracture elastic theory, The water inrush characteristics of the whole coal seam are divided into three "solid-liquid" coupling stages, the original gap seepage stage, the initial water discharge stage of mining fissures and the water inrush stage of fractured rock mass. The steady value of water inflow and its variation characteristics with time are predicted by using the formula of deep well flow in confined aquifer.

Point 2: Additionally, the authors need to address knowledge gaps on the studied research topic to depict the importance of the study.

Response 2: We appreciate the considerate comment. The first paragraph is the introduction,we  add the content of the current research situation to the paper and reflect the importance of the research.

Point 3: Need more explanation of all results from Table 1, how authors got the results in terms of experimental procedures.

Response 3: We appreciate the considerate comment. According to the construction drilling results and the laboratory physical and mechanical property test results of construction rock samples during mine field exploration (the test is based on DZ / T 0276 rock physical and mechanical property test specification), the statistical values of various parameters in Table 1 are obtained.

Reviewer 2 Report

research flowchart needed 

Author Response

Point 1: In the Abstract part:: It should be more focused on the problem statement, research methodology, and the main finding in this paper, therefore the abstract should be rewritten. the last sentence is double and should delete "The steady value of water inflow and its variation characteristics with time are predicted by using the formula of deep well flow with constant drop of confined aquifer to guide the formulation of prevention and control measures".

Response 1: We appreciate the considerate comment.We changed the abstract and deleted the duplicate parts.

In order to grasp the overburden evolution law of the shallow and short coal seam group, based on the key bed theory, the mechanical analysis of the longitudinal expansion of mining-induced overburden fractures in the coal seam group was carried out, and the spatial evolution characteristics of mining-induced overburden fractures were simulated by numerical simulation method. The results shown that in the process of working face advancement, with the continuous instability and failure of the overburden,the size and shape of fracture network are also changed. The repeated mining of the lower coal seam further causes the secondary activation of the upper overburden, which makes the roof fractures of the partially compacted goaf violently move again. The "channel source" and "space source" continue to carry out the process of "generation-expansion-compression-generation-expansion", in combination with pore fracture elastic theory, The water inrush characteristics of the whole coal seam are divided into three "solid-liquid" coupling stages, the original gap seepage stage, the initial water discharge stage of mining fissures and the water inrush stage of fractured rock mass. The steady value of water inflow and its variation characteristics with time are predicted by using the formula of deep well flow in confined aquifer.

Point 2: It seems there is no introduction for this paper, moreover introduction part should focus on a literature review about the topic. This part should be written.

Response 2: We appreciate the considerate comment. The first paragraph of the papers is the introduction, and the content of research status is added to the original text, and reflects the importance of research.

Point 3: The authors should present the research methodology approach(research methodology flowchart).

Response 3: We appreciate the considerate comment.We add the research methodology flowchart.

Point 4: In the research methodology part should mention about standards they used for the laboratory part(ASTM, BS, Uunfied.....)

Response 4: We appreciate the considerate comment. Test results of physical and mechanical properties in laboratory (according to DZ/T 0276 Rock physical and mechanical Properties Test code).

Point 5: The paper should be written very well in English to be understandable.

Response 5: We appreciate the considerate comment. Language errors have been corrected.

Point 6: There is no verification or validation or comparison for the results.

Response 6: We appreciate the considerate comment. The results of different ways of comparing appear in section 3.

Point 7: The topic is relevant to the special issue of the journal.

Response 7: We appreciate the considerate comment.

Point 8: The similarity check was attached(10%).

Response 8: We appreciate the considerate comment.

Point 9: The conclusions are consistent with the evidence and arguments presented

Response 9: We appreciate the considerate comment.

Reviewer 3 Report

The study is lack of references for the elaborating of the key scientific problems, such as the influence of long-term strength of rock mass, refering to 1) Shear behaviors of granite fractures immersed in chemical solutions, Engineering Geology, 2020, 279: 105869. 2) The role of water lubrication in critical state fault slip, Engineering Geology, 2020, 271: 105606. 3) Dou, Zihao; Shang, Delei; Zhao, Zhihong; Gao, Tianyang; Li, Jinjin; Yang, Qiang, Experimental study on water injection induced fault slip under critical state, 14th International Congress on Rock Mechanics and Rock Engineering, ISRM 2019, Foz do Iguaçu, Brazil, 2019-09-13 To 2019-09-18. The manuscript is organized like a report or academic dissertation, please clarify the relationship between the study and the others before being accepted for publication.

Author Response

Point 1: The study is lack of references for the elaborating of the key scientific problems, such as the influence of long-term strength of rock mass, refering to 1) Shear behaviors of granite fractures immersed in chemical solutions, Engineering Geology, 2020, 279: 105869. 2) The role of water lubrication in critical state fault slip, Engineering Geology, 2020, 271: 105606. 3) Dou, Zihao; Shang, Delei; Zhao, Zhihong; Gao, Tianyang; Li, Jinjin; Yang, Qiang, Experimental study on water injection induced fault slip under critical state, 14th International Congress on Rock Mechanics and Rock Engineering, ISRM 2019, Foz do Iguaçu, Brazil, 2019-09-13 To 2019-09-18. The manuscript is organized like a report or academic dissertation, please clarify the relationship between the study and the others before being accepted for publication.

Response 1: We appreciate the considerate comment and we carefully consulted and compared the three papers. The manuscript is independent and has no connection with these three papers. 1) the papers studied on the influence of different pH solutions on the mechanical strength of rock fractures. 2) This paper mainly studies the effect of water lubrication on fault slip by reducing the adhesion of rock mass surface. 3) This paper also focuses on the effect of water on fault slip.

Round 2

Reviewer 1 Report

The revised manuscript can be processed for further steps to publish.